# A Monte-Carlo/FDTD Study of High-Efficiency Optical Antennas for LED-Based Visible Light Communication

**DOI:** 10.3390/nano12203594

**Published:** 2022-10-13

**Authors:** Darya Fakhri, Farid Alidoust, Ali Rostami, Peyman Mirtaheri

**Affiliations:** 1OIC Research Group, Faculty of Electrical and Computer Engineering, University of Tabriz, Tabriz 5166614761, Iran; 2SP-EPT Laboratory, ASEPE Company, Industrial Park of Advanced Technologies, Tabriz 5364196795, Iran; 3Department of Mechanical, Electronics and Chemical Engineering, OsloMet—Oslo Metropolitan University, 0167 Oslo, Norway

**Keywords:** VLC, OWC, optical antenna, FDTD, ray tracing, monte-carlo, quantum-dot

## Abstract

In high-speed wireless communication, visible light communication is considered an emerging and cutting-edge technology. A light-emitting diode can serve both as an illumination source in an environment and as a data transmitter. Nevertheless, plenty of complications stand in the way of developing VLC technology, including the low response time of waveguides and detectors and the field of view dependence of such devices. To cover those challenges, one approach is to develop a superior optical antenna that does not have a low response time related to phosphorescence materials and should also support concentrating light from the surroundings with a wide field of view. This research paper presents an optimized cylindrical optical antenna with benefits, such as affordable cost, fast response time due to high-efficient nanomaterials, and a wide field of view (FOV). The proposed structure avoids the need for intricate tracking systems and active pointing to the source, but it can also be integrated into portable devices. For the analysis of nanomaterials’ characteristics, finite difference time domain simulations are used, and Monte-Carlo raytracing is used to study the proposed optical antenna. It was found that the antenna’s optical efficiency varies from 1 to 29% depending on the size and the number of nanomaterials inside. Compared to other works, this paper shows higher efficiencies and wider FOV.

## 1. Introduction

With the rapid development of various digital technologies and the growing demand for high data rates in recent years, wireless communication plays an essential role in telecommunication equipment [1]. Before discussing in detail what happened in the past for this technology, it should be mentioned that VLC is a relatively new field, and only a few papers and reports have been published. In any case, we will address all challenges in this section.

This technology utilizes electromagnetic (EM) waves as communication media [2]. Radio Frequency (RF) communication, as a portion of wireless communication, has substantiated the growing demand for high data rates and greater bandwidth due to its bandwidth restriction with increasing network traffic [3,4,5,6,7,8].

A solution to this limitation was introduced by developing optical wireless communication (OWC) as an alternative technology to radio frequency communication, especially for indoor communications. VLC (visible light communication) is considered a new wireless communication technology capable of providing high data rates for indoor and outdoor applications. This system competes with fifth-generation (5G) radio frequency (RF) systems [2,9]. In OWC, the data carrier is light waves in the electromagnetic (EM) spectrum; hence, thanks to its broad bandwidth (300 GHz to 30 PHz), high data rate, high security, low cost, and high energy efficiency, it is a promising technology in wireless communications [2,6,8,9,10,11,12,13].

Visible light (VL) [14], ultraviolet (UV) [15], and infrared radiation (IR) [16] are the optical bands used as the propagation medium in OWC technology [2]. VLC (Visible Light Communication) is one of the most promising areas of OWC technology that utilizes the spectrum of visible light in the electromagnetic spectrum (400–800 THz) as a way to carry data as a carrier wave [6,14,17,18,19,20,21,22,23].

Light Emitting Diodes (LEDs), due to their high switching rate, are used as a suitable source in VLC, which transmit data in a fast and imperceptible way [2,24,25]. Moreover, LEDs, due to the reduction in energy consumption by about 80% [9], high efficiency, and long lifetime, are a revolutionary alternative to fluorescent lights and incandescent bulbs in lighting systems [23,26]. Since VLC can combine illumination with communications, there has been an increase in interest in this technology since its development.

Increasing the VLC channel’s capacity in the future is essential to meet users’ future demands and achieve higher bandwidth. Similar to many other communication systems, the capacity of a VLC channel is determined in part by the signal-to-noise ratio (SNR) and the channel bandwidth at the transmitter and receiver [3]. Therefore, a powerful signal must be received at the receiver to increase the channel capacity. For this purpose, the light should be collected by a large area and concentrated on the photodetector.

However, in the optical elements such as compound parabolic reflectors (CPC) or lenses, due to the conservation of etendue [3], their Field Of View (FOV) is limited and so is incompatible with mobile devices such as laptop computers and smart mobile terminals [27,28,29].

Luminescent solar concentrators (LSCs) are a solution to concentrate a considerable signal without detrimental effects on FOV. An LSC typically consists of a fluorescent material sandwiched by a cladding material such as glass or plastic [30,31,32,33]. In LSCs, fluorescent material changes the wavelength of photons by the Stokes shift. The shift in wavelength of the photons causes the system not to be based exclusively on refraction and reflection of light and, therefore, exceeds the etendue limit for optical gain [3].

Antennas based on LSC technology are constructed by absorbing irradiant light, which can be directed onto a large surface of the antenna and emitting it at a longer wavelength after being absorbed by the fluorescent material. As a result of the higher refractive index of glass or plastic than that of its surrounding environment (air), a portion of the emitted photons has reflected on the edge of the antenna by total internal reflection (TIR), where the photodetector is mounted to detect the photons emitted [34].

In this field, Manousiadis et al., in 2016, sandwiched a layer of Coumarin6 (Cm6) as the fluorescent material by glass coating with rectangular cube geometry and reported the optical gain equal to 12 with a field of view of ±60° [3]. In another report, in 2017, Dong et al. designed an LSC-based optical antenna with Super Yellow as the fluorescent material with a flat CPC structure and reported double optical efficiency compared to its rectangular counterpart [5]. In the most recent research, in 2022, Chamani et al. designed a fluorescent antenna with a narrow-edge cubic structure based on Graphene Quantum Dot (GQDs) with a quantum yield equal to 0.99. Finally, they obtained an optical efficiency equal to 1.058% [35].

Although LSCs are suitable for concentrating a significant part of incident light without reducing the FOV, they do have problems, as follows:

First, fluorescent materials have a high relaxation time, so they cannot switch high frequencies; hence, they are unsuitable for Light Fidelity (Li-Fi) applications. Second, most fluorescent materials are unstable due to the presence of sulfur and phosphorus in their composition and react quickly with air and water, hence, they must have a protective layer, which makes the fabrication process complicated and expensive. Third, due to the Stokes shift caused by the fluorescent material in these antennas, the output photon wavelength is always longer than the input photon wavelength. Thus, part of the input data does not appear in the output.

In addition to the problems mentioned for fluorescent antennas that have been designed so far, due to their non-circular structure, they do not show the same function for photons that hit them from different directions, and this problem has the potential to lose some photons and, thus, reduces optical efficiency. It should be noted that antennas with non-circular cross-sections do not couple well with photodetectors with circular active areas, such as APD detectors; hence, some photons in the area between the concentrator and the photodetector are lost due to the mismatch of their active surfaces.

The problems mentioned earlier motivated us to develop a new and promising optical antenna with high optical efficiency without the field of view limitation, overall sensitivity, good coupling capabilities with photodetectors (with circular active areas), long lifetime, and low cost. 

In this paper, our approach is to use a cylindrical glass structure doped with SiO_2_/Si nanoparticles. In our structure, we used this nanoparticle due to its significant characteristics, such as low relaxation times, which made it possible to use a high switching rate compared to fluorescent materials, high stability, and adjustable absorption spectra with its size, a considerable extinction cross-section, and its low cost. All of these factors attracted our attention, so we decided to use them as a luminescent material in our structure.

Moreover, the cylindrical geometry of the proposed structure has some significant advantages compared to cubic geometries, such as wide FOV, reasonable sensitivity, and excellent coupling with photodetectors; therefore, it causes the minimum losses in photons hitting the structure and high optical efficiency. This structure is designed to trap as much light as possible in the nanoparticles, so the light is guided by the glass cladding to the photodetector edges.

We have also optimized the dimensions of the nanoparticle to ensure maximum overlap between the emission spectrum of the light source and the absorption spectrum of the nanoparticle, especially in peak regions of the spectrum. This will reduce losses for the nanoparticle to absorb and re-emit all the input light data.

We used the Monte-Carlo ray tracing method for simulating the optical antenna structure and the finite-difference time-domain (FDTD) method to obtain the desired absorption and emission spectra of the SiO_2_/Si nanoparticle and eventually reach the desired structure for an optical antenna.

## 2. Materials and Methods

### 2.1. Antenna Structure and Underlying Physics

A glass cylinder embedded with core-shell Silicon Dioxide–silicon nanoparticles is subjected to white LED radiation in the proposed structure. A white LED usually contains a large band-gap material that emits the blue region of the visible spectrum and a layer of phosphorescence that acts as a down converter material such as Ce: YAG or phosphor. It should mention that the added QDs act as plasmonic nanoparticles, and their optical properties differ from those of Si QDs alone.

These LEDs have a peak emission of 450 nm, and the phosphorescence material down-converts some part of these photons to longer wavelengths around 550 nm (Figure 1).

Generally, in this structure, SiO_2_/Si nanoparticles absorb the incident light from the cylinder’s lateral surface and then emit it respectively. The photons emitted by the nanoparticles (through the TIR phenomenon) will be able to reach the edges of the cylinder where the photodetectors are located due to their change in the mean free path length of the incident photons. Finally, the absorbed photons by photodetectors are converted into an electrical signal (Figure 2).

It should be mentioned that high-speed communication within VLC is directly related to the fast response of the materials used in the equipment. Although direct band-gap materials, such as GaAs, can achieve higher frequencies in VLCs, it is important to remember that VLCs have been developed to cover both broad bandwidth and low cost. In consumer-level VLC applications, using direct band-gap materials is not economical. Despite this, the decay time of SiO_2_/Si QDs is in the nanosecond range, which could provide the bandwidth required for VLCs.

In this structure, different events for each photon can occur. Since two media (air and glass) have different refractive indices, the incident photon may be reflected from the surface of the cylinder in the first place. If the photon is not reflected, it enters the structure and may be absorbed by the nanoparticle or hit the surfaces and the phenomenon of total internal reflection occurs or pass through the structure. The absorbed photon by the nanoparticle may not be emitted; thus, the absorption loss happens. If the nanoparticle emits the photon, one of the following three phenomena will occur: First, the photon hits the structure surfaces at an angle smaller than the critical angle, so it escapes. Two, the photon hits the structure surfaces at an angle greater than the critical angle, so the TIR phenomenon occurs. Three, the photon is re-absorbed by the other nanoparticle, for which this mechanism is known as re-absorption.

Different events in Figure 3 for incident photons can be described as follows: [37]

(1)The photon passes through the cylinder without being absorbed by the nanoparticles (transmission losses).(2)The photon is absorbed by the nanoparticle and then is emitted and escapes from the cylinder because its incident angle with the surface is smaller than the critical angle (transmission losses).(3)The photon is absorbed by the nanoparticle and is emitted and then absorbed by another nanoparticle (re-absorption), and (3, 6) is not emitted (absorption losses). To be more precise, each photon’s absorption loss can be calculated using Equation (10).(4)The photon is absorbed by the nanoparticles and emitted and then reaches the photodetector by the TIR phenomenon.(5)The photon is reflected from the surface of the cylinder without entering it.

### 2.2. Simulation

#### 2.2.1. FDTD Simulation

To calculate the absorption and emission spectra of the nanoparticle, we placed the SiO_2_/Si nanoparticle in a medium with a background refractive index of SiO_2_ (1.46) and exposed it to planar source radiation ranging from 300 to 800 nanometers. Then, we obtained the nanoparticle absorption and emission spectra for its different dimensions employing the FDTD (finite-difference time-domain) method. Figure 4 and Table 1 represent the FDTD region and its related parameters, respectively. 

As shown in Figure 4, several areas in an FDTD simulation analyze the absorption and scattering curves of a nanoparticle. In the list below, one can find an explanation of each area in Figure 4. Table 1 also describes the parameters used in the FDTD analysis, including their dimensions.Boundary conditions of the FDTD region,Background medium,Scattering calculation region,Planar light source,Absorption calculation region,Shell material, andCore material.


#### 2.2.2. Monte Carlo Simulation

In general, 100,000 photons are emitted from the LED to the antenna in the range of $\left[-\theta_{T},\theta_{T}\right]$ is the angle of incidence of the LED, which is shown in Figure 5 and presented in Equation (1). As the antenna’s length increases, the antenna will be able to receive more photons within its acceptance angle (Figure 5, Equation (2)):(1)$$\theta_{T}=\tan^{-1}\left(\frac{\frac{1}{2}\times h_{T}}{d}\right)$$
where *d* is the orthogonal distance from the LED to the antenna and *h_T_* is the maximum range where photons can travel.
(2)$$\theta_{T}=\tan^{-1}\left(\frac{\frac{1}{2}\times h_{T}}{d}\right)$$

In Equation (2), *L* represents the length of the antenna.

In this work, *d* and *h_T_* are considered 100 and 10 cm, respectively. Thus, according to Equation (2), *θ_T_* is equal to 2.86°. In addition, we changed the antenna’s length from 2 to 10 cm and the antenna’s radius from 1 to 5 cm with a step of 2 cm.

Simple integration and averaging cannot be used to calculate the optical efficiency of the proposed structure due to the reabsorption mechanism and probabilistic events. In applied mathematics and engineering, a Monte-Carlo technique is an approach that generates random numbers in an attempt to solve a problem. It can be used in cases where a deterministic algorithm cannot be used or where the variables of the problem have coupled degrees of freedom [37,38,39].

The Monte-Carlo ray tracing process determines a photon’s ultimate fate based on empirical data and mathematical equations, including a photon’s absorption and emission spectrum, quantum yield, Snell’s law, Beer–Lambert’s law, and other related factors [40]. In an illustration of the ultimate fate of each photon, Figure 6 illustrates the process flow in this simulation. Upon completing the algorithm, we can determine the fate of all photons and conclude the overall structure’s optical efficiency.

The simulation’s first step is generating photons with a specific angle and wavelength. The angle of each photon is selected randomly in the range of $\left[-\theta_{T},\theta_{T}\right]$ and also the photons with angles in the range of $\left[-\theta_{T},\theta_{T}\right]$ can hit the antenna. We can determine every photon’s wavelength by converting the white LED spectrum *PDF* to the cumulative distribution function (*CDF*) and then using inverse transform sampling [41].

Equation (3) shows how to calculate *PDF* [37].
(3)$$PDF=\frac{Distribution\;Function\;of\;the\;Curve}{Area\;of\;the\;Curve}$$

Moreover, the *CDF* of the *i*th term is defined as the sum of the *PDF*s to the *i*th term divided by the area under the *PDF* curve calculated by the trapezoidal method (Equation (4)). Here *λ_j_* is the wavelength of the *j*th photon, and k is its last term of the series [37].
(4)$$CDF_{i}=\frac{\sum_{j=1}^{i}\frac{\left(PDF_{j+1}\;+\;PDF_{j})(\lambda_{j+1}\;-\;\lambda_{j}\right)}{2}}{\sum_{j=1}^{k}\frac{\left(PDF_{j+1}\;+\;PDF_{j})(\lambda_{j+1}\;-\;\lambda_{j}\right)}{2}}$$

Figure 7A,B represents the *PDF* and *CDF* of the white LED, respectively.

Each photon strikes the structure’s surface at a random angle $\left[-\theta_{i},\theta_{i}\right]$. Based on Snell’s law [42], a reflected photon will not participate in the following steps of Monte Carlo Analysis. Due to the re-emission of photons from the surface of the object, this is known as a reflection loss. Using Equations (5) and (6), Fresnel reflectance is calculated for vertical and parallel polarized light [43]. In the case of s-polarized light, the reflectance is as follows:(5)$$R_{s}={\left|\frac{n_{1}\cos\theta_{i}\;-\;n_{2}\cos\theta_{t}}{n_{1}\cos\theta_{i}\;+\;n_{2}\cos\theta_{t}}\right|}^{2}=\;{\left|\frac{n_{1}\cos\theta_{i}\;-\;n_{2}\sqrt{1-{\left(\frac{n_{1}}{n_{2}}\sin\theta_{i}\right)}^{2}}}{n_{1}\cos\theta_{i}\;+\;n_{2}\sqrt{1-{\left(\frac{n_{1}}{n_{2}}\sin\theta_{i}\right)}^{2}}}\right|}^{2}$$

The refractive index of the waveguide (*n*_2_) is considered constant and equal to the refractive index of SiO_2_ since *n*_1_ = *n*_Air_ = 1, and the concentration of the nanoparticles is not high enough to change it considerably. The *θ_i_* is the angle of the incident photon on the structure, and *θ_t_* is its transmission angle. In the case of p-polarized light, the reflectance is as follows:(6)$$R_{p}={\left|\frac{n_{1}\cos\theta_{t}\;-\;n_{2}\cos\theta_{i}}{n_{1}\cos\theta_{t}\;-n_{2}\cos\theta_{i}}\right|}^{2}=\;{\left|\frac{n_{1}\sqrt{1-{\left(\frac{n_{1}}{n_{2}}\sin\theta_{i}\right)}^{2}}-n_{2}\cos\theta_{i}\;\;}{n_{1}\sqrt{1-{\left(\frac{n_{1}}{n_{2}}\sin\theta_{i}\right)}^{2}}+\;n_{2}\cos\theta_{i}\;}\right|}^{2}$$

Incident light can be considered unpolarized, so the surface’s reflection can be considered an equal mix of *s* and *p* polarizations (Equation (7)) [44].
(7)$$R=\frac{1}{2}(R_{s}+R_{p})$$

Upon entering the structure, the photon’s position can be determined by calculating its angle of transmission (*θ_t_*), and distance traveled. *θ_t_* is determined using Snell’s law (Equation (8)) [42]. Earlier, we introduced *n*_1_ and *n*_2_.
(8)$$\frac{\sin\theta_{i}}{\sin\theta_{t}}=\frac{n_{2}}{n_{1}}$$

To figure out how far a photon travels, the Beer–Lambert law is used. By applying this law, we can determine a ratio of the likelihood of a photon being absorbed via the absorption pathway (Fractional absorbance (*A*)) [37,45,46,47,48,49].
(9)$$A=1-10^{-\;\alpha (\lambda )\Delta \lambda }$$

Moreover,
(10)α(λ) = ε(λ) c

In this case, *ε*(*λ*) is the wavelength-dependent absorption coefficient of the SiO_2_/Si nanoparticles, *c* is the concentration of the so-called material, and Δ*L* is the path length traveled by the photon before being absorbed. Please refer to Figure 8 for *ε*(*λ*).

Moreover,
(11)$$\Delta L\;=\;-\;\frac{\log(1-A)}{\alpha (\lambda )}$$

In this simulation, (*γ* = 1 − *A)* corresponds to a random number ranging from 0 to 1, so the distance traveled by the photon can be calculated using the Equation (12). This allows us to determine the position of the photon.
(12)$$\Delta d\;=\;-\;\frac{\log(\gamma )}{\alpha (\lambda )}$$

According to the photon’s wavelength and position, it is determined whether the nanoparticles absorb the photon or not. If the photon’s wavelength is not in the range of the absorption spectrum of the nanoparticle, it will pass through the structure without being absorbed, known as transmission losses. Otherwise, the photon’s position determines whether the nanoparticles absorb the photon; if the photon’s position is inside the cylinder, the photon is absorbed by the nanoparticle. Otherwise, it has interacted with the structure’s surface. Two scenarios can occur in this case: first, the photon is reflected from the surface by TIR, then, it is checked whether it is absorbed by nanoparticles or interacts with surfaces again. If the photon escapes from face 2 or 3 (F_2_ or F_3_) (Figure 5), it is harvested by the photodetector; second, the photon escapes from face one, which is known as transmission loss.

An essential step in this process involves determining whether or not the nanoparticle emits the absorbed photon. A nanoparticle’s probability of emitting photons can be determined by the quantum yield (QY) of the nanoparticle, which is defined as the ratio of the number of photons emitted to the number of photons absorbed (Equation (13)).
QY = Emitted Photons/Absorbed Photons(13)

The simulation generates a random number (*R*) between 0 and 1 compared with QY. If the random number is smaller than QY, the nanoparticle emits the photon, and the photon’s propagation angle is random (this is because Rayleigh scattering occurs when the particle size is smaller concerning the wavelength of the incident light source). The re-emitted photon wavelength is obtained from the CDF of the nanoparticle’s emission spectrum (Figure 9A,B represent the PDF and CDF of the nanoparticle’s emission spectrum, respectively), and, hence, its traveled distance can be calculated using Equation (14).
(14)$$\Delta d\;=\;-\frac{1}{\varepsilon (\lambda )c}\log R$$

If the (*R*) value is greater than the QY, the photon is not emitted, which is known as re-absorption losses.

If the photon is emitted from the nanoparticle, its new position is stored, and the previous steps are repeated. Finally, optical efficiency (*η_opt_*) is defined as the ratio of photons collected from *F_2_* and *F_3_* to all photons emitted by the *LED* (Equation (15)).
(15)$$\eta_{opt}=\frac{Collected\;photons\;from\;F2\;+\;Collected\;photons\;from\;F3\;}{photons\;emitted\;from\;LED}$$

## 3. Results

As we said before, to realize the most appropriate absorption and emission spectra for the nanoparticle, we have analyzed the size of the nanoparticle by the FDTD method. According to Figure 10, Figure 11 and Figure 12, the absorption, scattering, and extinction cross-sections of SiO_2_/Si nanoparticles are shown for the core thickness of 6 nanometers and the shell thickness of 75 to 95 nanometers, respectively.

Figure 11 presents the Scattering coefficient, whereas Figure 12 displays the Extinction coefficient versus wavelength for SiO_2_/Si nanoparticles of different sizes. In these figures, the peaks represent the local surface plasmon resonances (LSPRs) that occur between SiO_2_ and Si.

According to Figure 10, Figure 11 and Figure 12, we selected the SiO_2_/Si nanoparticle with a radius of 85 nm because the peak of the extinction cross-section occurs in the two places (450 and approximately 550 nm), matching the emission spectrum of the white LED (Figure 1). The absorption cross-section of the nanoparticle with a radius of 85 nm is shown in Figure 13.

By comparing Figure 1 and Figure 13, the absorption spectrum of the nanoparticle has wholly overlapped in the frequency domain, and the peak points of the nanoparticle’s absorption spectrum are a significant match with the peak points of the LED’s emission spectrum. Therefore, this nanoparticle has all the conditions necessary to absorb and emit the LED’s wavelengths.

### 3.1. CIE Colorspace Comparison between LED Illumination and SiO_2_/Si QD Scattering

In the CIE-1931 color space, the distributions of wavelengths in the visible electromagnetic spectrum are quantitatively linked to colors as humans perceive them. In color management, which includes issues such as displays, cameras, and, in our case, optical antennas, there are mathematical relationships that define the distinct color spaces in CIE-1931 format. It is necessary to have mathematical relationships for color management to function effectively and for optical antennas to be highly efficient.

A standard observer’s chromatic response is modeled using the CIE color-space by mapping wavelength power spectra to an ensemble of stimulus values, *X*, *Y*, and *Z*, representing the actual response of the three types of cone cells in the eye. Equation (16) shows the relationships between the chromatic response of the material and the different values that *X*, *Y*, and *Z* could get and Figure 14 shows wavelength dependency of *X*, *Y*, and *Z*.
(16)$$X=\int P\left(\lambda \right)\bar{x}\left(\lambda \right)d\lambda \;Y=\int P\left(\lambda \right)\bar{y}\left(\lambda \right)d\lambda \;Z=\int P\left(\lambda \right)\bar{z}\left(\lambda \right)d\lambda$$

With Equation (17), it is possible to normalize the *X*, *Y*, and *Z* values (although it may lose information regarding the brightness (amplitude) of the light).
(17)$$x=\frac{X}{X+Y+Z},\;\;y=\frac{Y}{X+Y+Z},\;\;z=\frac{Z}{X+Y+Z}=1-x-y$$

Therefore, two parameters can be used to describe the color of light, *x*, and *y*.

Figure 15 illustrates the chromaticity diagram based on the CIE standard, illustrating the spectrum of white LED (from Figure 1) and that of a proposed SiO_2_/Si Quantum Dot with a radius of 85 nm. It is evident in Figure 15 that both the transmitter (white LED) and the receiver (SiO_2_/Si QDs inside glass substrate) have similar color representations, and this can aid in constructing an optical antenna with greater efficiency.

### 3.2. Results for Monte-Carlo Ray Tracing

In a Monte-Carlo ray-tracing simulation, waveguide and surrounding medium refractive indices, absorption and emission spectra of nanoparticles, waveguide dimensions, nanoparticle concentration, and quantum yield are input parameters. The quantum yields of nanoparticles can vary between zero and one, depending on their material. During this study, the QY value of the SiO_2_/Si nanoparticles was considered, at values of 0.3, 0.6, and 0.95, to be as comprehensive as possible.

In the presented structure, to have high optical efficiency, we have calculated the optical efficiency according to Equation (15) in different dimensions of the structure with various concentrations of nanoparticles. To ensure the accuracy of the data regarding efficiency, we carried out these simulations repeatedly using the Monte-Carlo method. Therefore, each reported efficiency is a mean of the results for five consecutive runs of the simulation corresponding to that efficiency. In Figure 16, we show optical efficiency for the structure with different dimensions using distinct concentrations of nanoparticles. In this figure, the results are reported for the various quantum yields mentioned earlier.

It is evident from Figure 16 that optical efficiency is reduced at both low and high concentrations. (It is important to note that the concentrations in this figure are on a logarithmic scale.) It can be said that the reduction in optical efficiency in high concentrations is due to absorption losses and, in low concentrations, due to transmission losses.

This research aims to find optimal concentrations, and from Table 2, Table 3 and Table 4, anyone can find optimal concentrations and optical efficiency for structures with radii of 1, 3, and 5 cm and different lengths at quantum yields of 0.3, 0.6, and 0.95.

Based on the obtained results, optical efficiencies over 20% are obtained at radii of 3 and 5 cm with a quantum yield of 0.95. To minimize the mismatch between the cross-section area of the proposed structure and that of photodetectors, low-radius structures are more accessible to couple to detector devices such as APDs. To achieve high efficiency, we recommend fabricating an optical antenna with a cylindrical structure with a radius of 3 cm and a length of 10 cm doped with the nanoparticles mentioned above.

For efficiencies greater than 20 percent and less than 1 percent, Figure 15 and Figure 16 illustrate the number of photons exiting from the edges of the cylinder at each wavelength.

It is evident from Figure 15 and Figure 16 that in optical efficiencies exceeding 29 (%), all data from the input spectrum (white LED) appear at the antenna output. As a result, we can reconstruct the signal sent by the LED with higher SNR (Figure 17). Nevertheless, at low optical efficiencies (less than 1 (%)), it is not guaranteed that all spectrum data will appear on the antenna output, which can make recovering the modulated signal difficult (Figure 18).

## 4. Conclusions

This work aimed to demonstrate a new structure for optical antennas for visible light communication (VLC) usage. There should be attributes such as simplicity, speed (short response time), sensitivity in all directions of incidence, relatively small size, and low price. Due to its proposed structure, this antenna is designed with a large field of view and a high signal-to-noise ratio. In this antenna, the host is a cylindrical glass substrate that is doped with specific Quantum Dots of SiO_2_/Si. An FDTD analysis was conducted on SiO_2_/Si quantum dots to determine their optimum size to be used as dopants inside the cylindrical substrate. An analysis of the absorption, scattering, and extinction cross sections of SiO_2_/Si QDs was carried out using the FDTD method. An optimal radius of 79 nm was determined for SiO_2_/Si nanoparticles that match the spectrum of source white LEDs. The SiO_2_/Si nanoparticle with this size shows absorption, scattering, and extinction cross sections of 6.65 × 10^−14^ m^−2^, 4.4 × 10^−13^ m^−2^, and 5.05 × 10^−13^ m^−2^. We numerically modeled the proposed optical antenna using the Monte-Carlo ray-tracing approach, and we reported the optical efficiency for a variety of substrate sizes and dopant concentrations inside the substrate.

Furthermore, the optical efficiency for the proposed structure was found to be in the range of 1 to 29% for various sizes and concentrations of dopants. The antenna substrate is doped with efficient SiO_2_/Si Quantum dots, which have a low relaxation time compared to phosphorescence-based LSCs, so that it could be applied to VLC applications demanding fast response times. A cylindrical surface and a wide field of view make it an excellent light-collecting antenna, liberating a VLC system from active light-tracking systems.

For future work, we will use different quantum dots to achieve better antenna performance for different visible band wavelengths. In this way, multiple users in a nearby area could use VLC with the antenna via wavelength division multiplexing.

## Figures and Tables

**Figure 1 nanomaterials-12-03594-f001:**
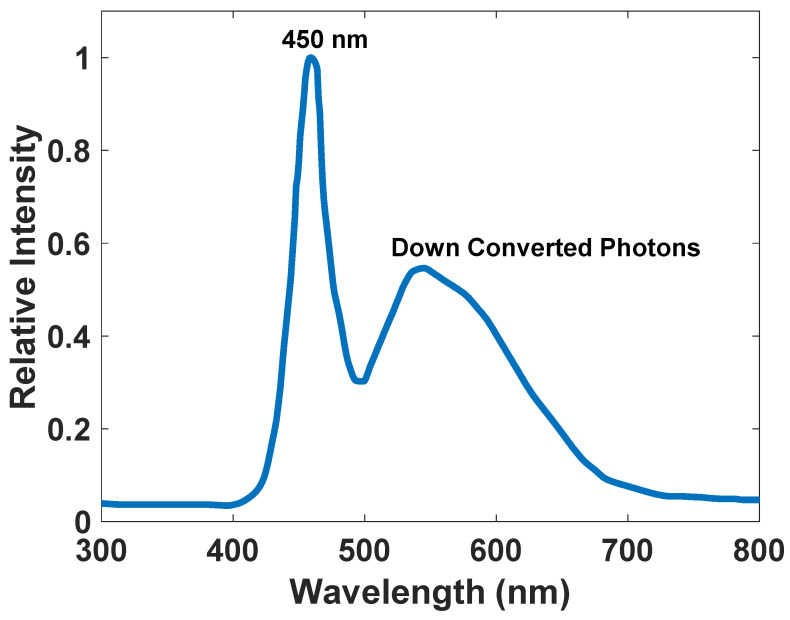
The emission spectrum of a commercially available white light LED. Adapted from Ref [36].

**Figure 2 nanomaterials-12-03594-f002:**
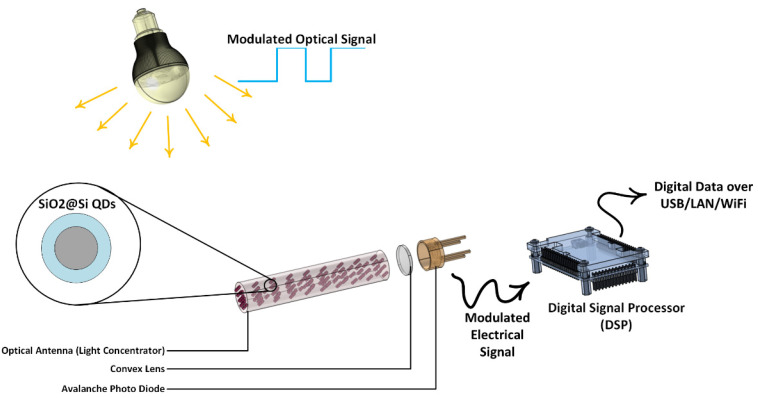
Schematic description of the proposed optical antenna in a VLC Setup.

**Figure 3 nanomaterials-12-03594-f003:**
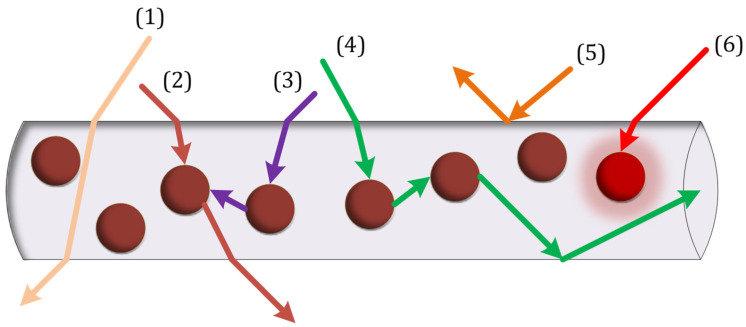
Physical phenomena in the proposed structure.

**Figure 4 nanomaterials-12-03594-f004:**
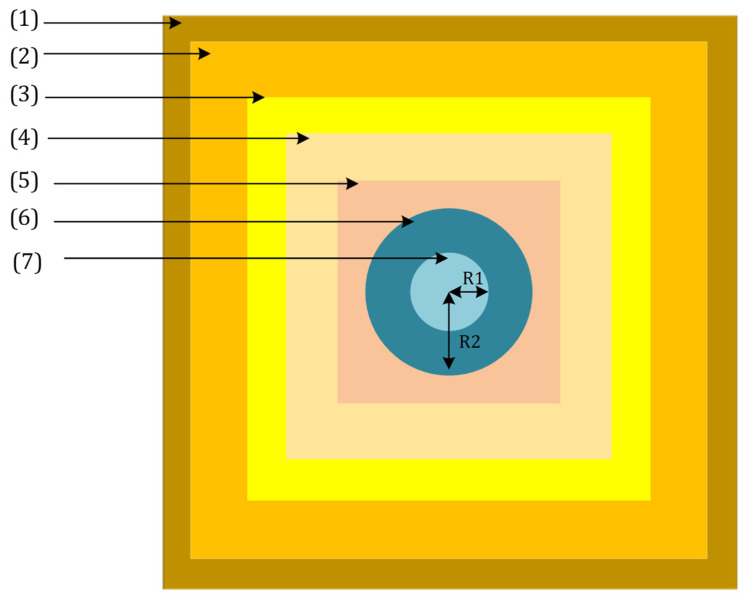
FDTD simulation environment for a nanoparticle.

**Figure 5 nanomaterials-12-03594-f005:**
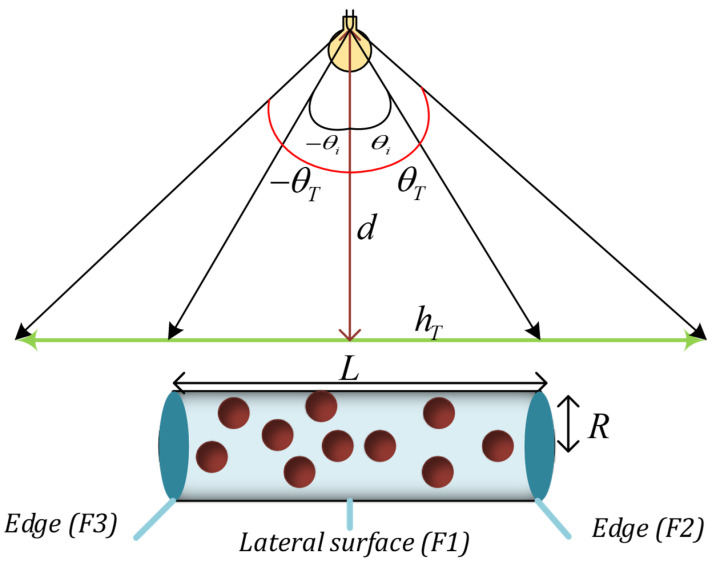
A schematic of how LEDs radiate to the optical antenna.

**Figure 6 nanomaterials-12-03594-f006:**
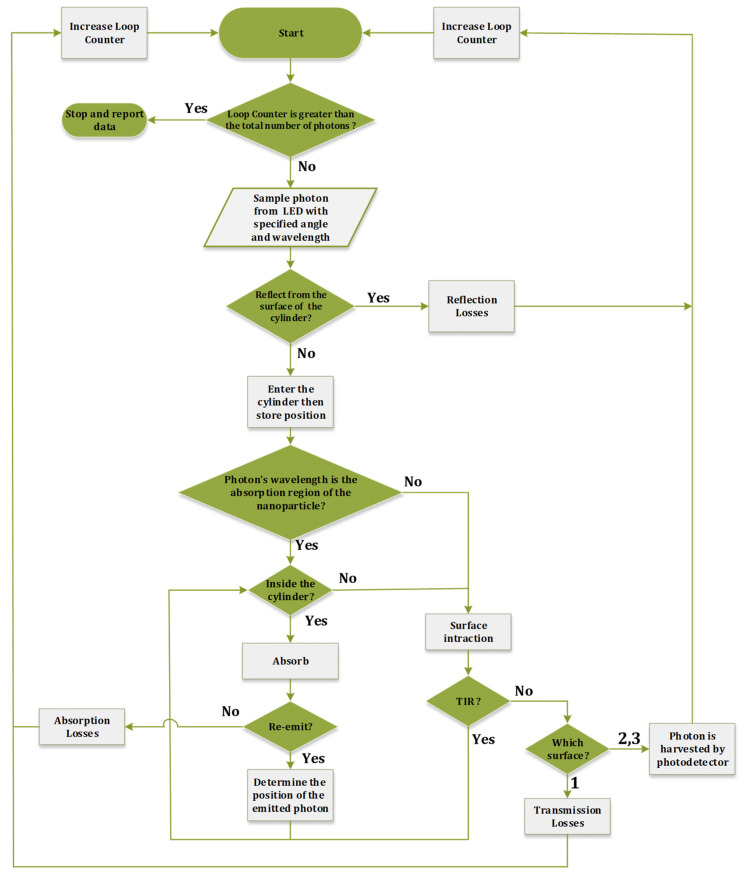
Algorithm of Monte-Carlo ray-tracing for the proposed structure.

**Figure 7 nanomaterials-12-03594-f007:**
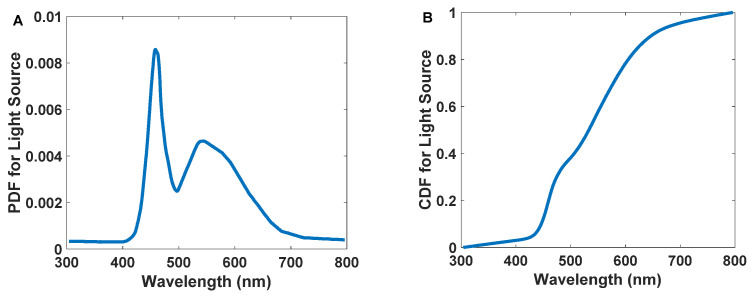
The probability density function (**A**) and cumulative distribution function (**B**) for commercially available white light LED emission spectrums.

**Figure 8 nanomaterials-12-03594-f008:**
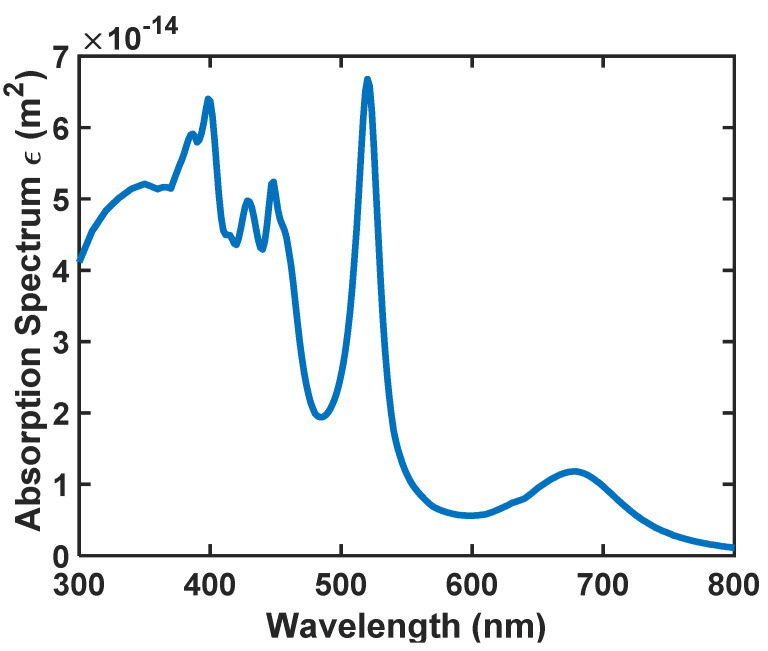
The absorption spectrum of the nanoparticle’s SiO_2_/Si with *R* = 85 nm.

**Figure 9 nanomaterials-12-03594-f009:**
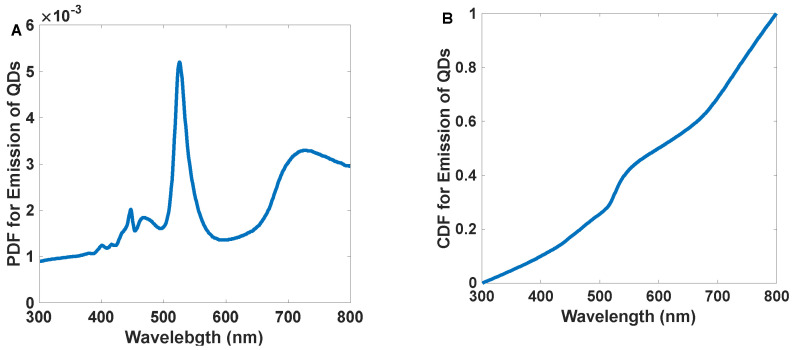
Probability density function (PDF) and cumulative distribution function (CDF) for emission spectrum of core-shell SiO_2_-silicon nanoparticle (**A**,**B**), respectively.

**Figure 10 nanomaterials-12-03594-f010:**
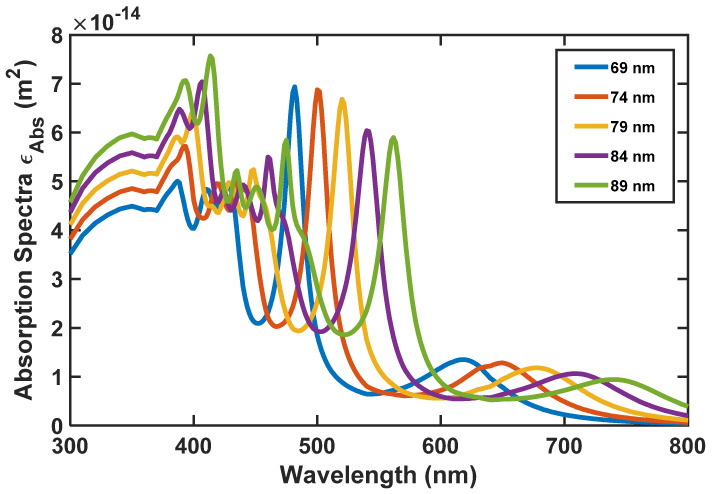
SiO_2_/Si nanoparticle’s absorption spectra for its different radii.

**Figure 11 nanomaterials-12-03594-f011:**
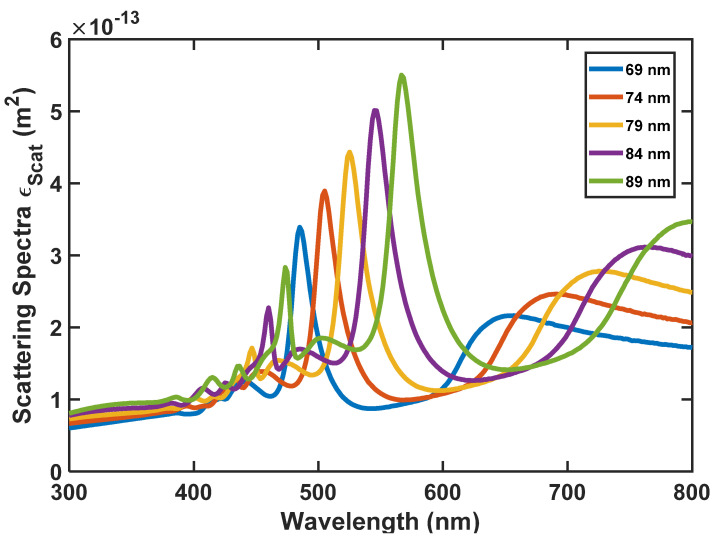
SiO_2_/Si nanoparticle’s scattering spectra for its different radii.

**Figure 12 nanomaterials-12-03594-f012:**
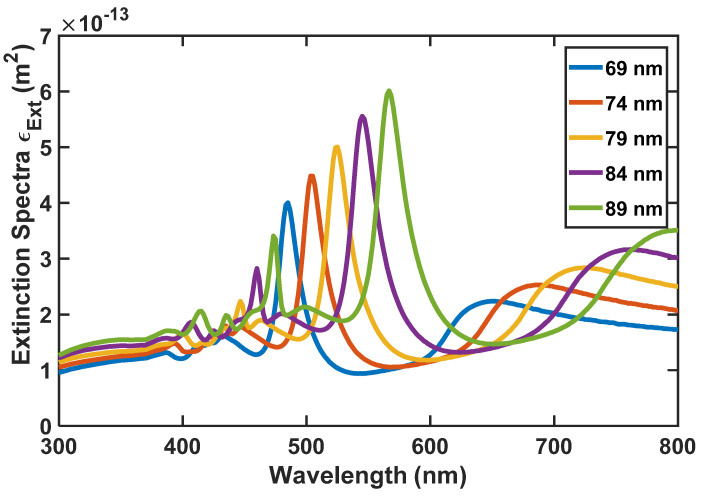
SiO_2_/Si nanoparticle’s extinction spectra for its different radii.

**Figure 13 nanomaterials-12-03594-f013:**
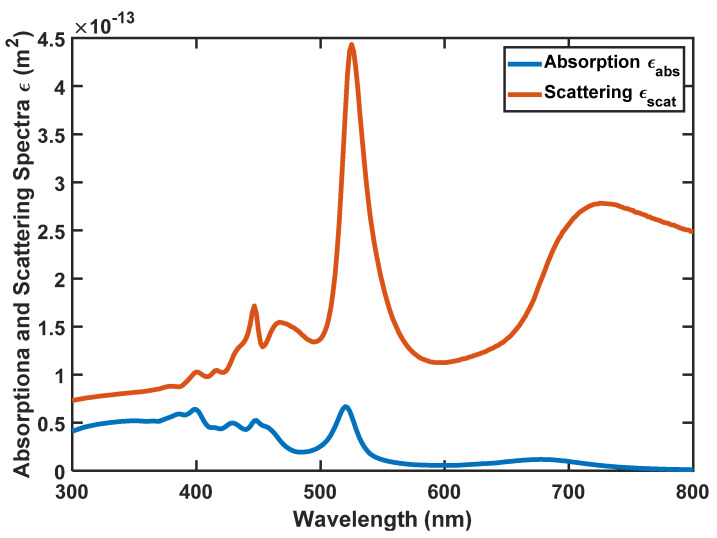
Absorption and scattering spectra of SiO_2_/Si nanoparticle with *R* = 85 nm.

**Figure 14 nanomaterials-12-03594-f014:**
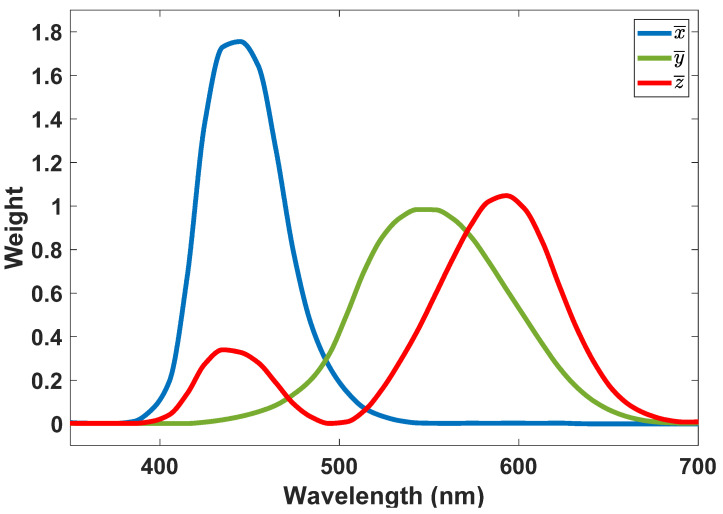
CIE 1931 profiles for blue, green, and red channels.

**Figure 15 nanomaterials-12-03594-f015:**
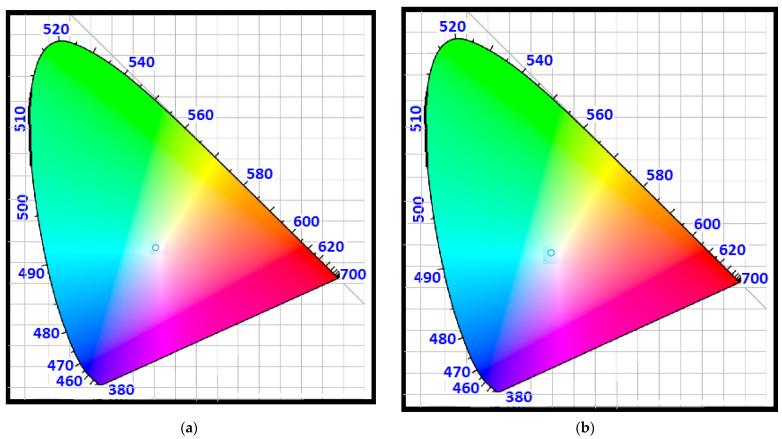
CIE 1931 representations for (**a**) white LED from Figure 1 and (**b**) proposed SiO_2_/Si nano-particle with *R* = 85 nm.

**Figure 16 nanomaterials-12-03594-f016:**
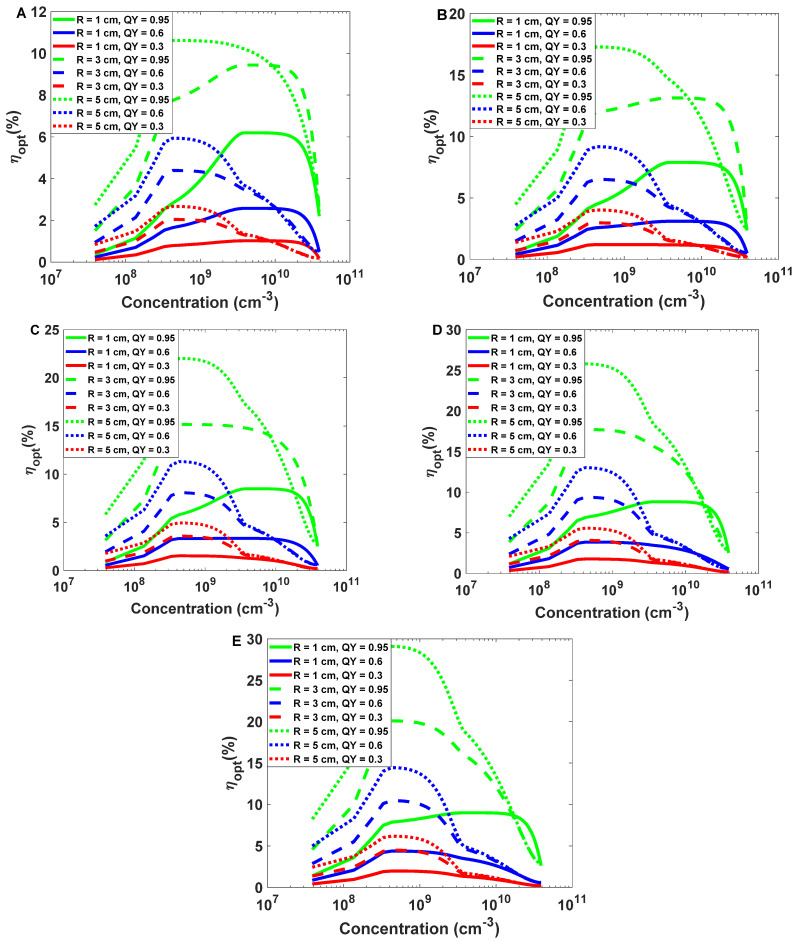
Optical efficiency concerning concentrations of nanoparticles in radii of 1, 3, and 5 cm with lengths of 2, 4, 6, 8, and 10 cm (**A**–**E**), respectively.

**Figure 17 nanomaterials-12-03594-f017:**
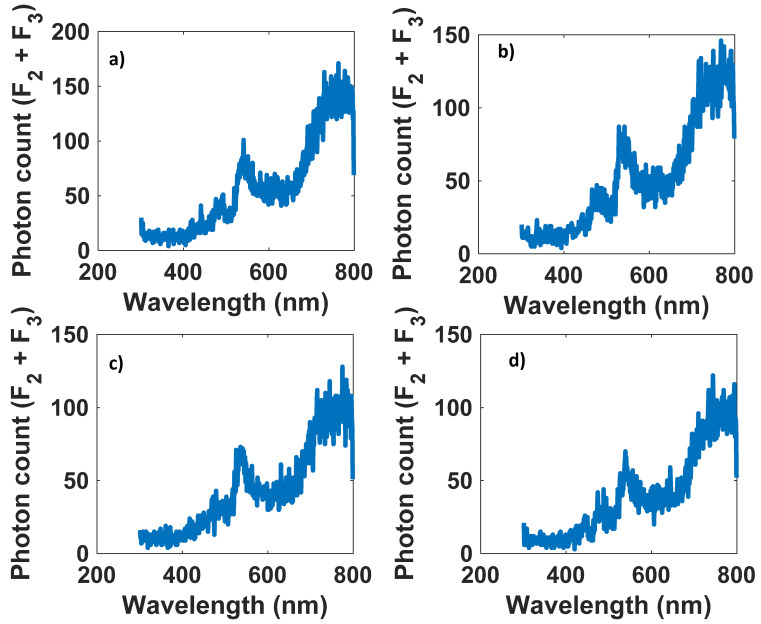
Structures with optical efficiencies over 20 (%). (**a**) L = 10 cm, *R* = 5 cm, QY = 0.95, Concentration = 3.88 × 10^8^ (1/cm^3^), *η_opt_* = 29.0990 (%). (**b**) L = 8 cm, *R* = 5 cm, QY = 0.95, Concentration = 3.88 × 10^8^ (1/cm^3^), *η_opt_* = 25.7964 (%).(**c**) L = 6 cm, R = 5 cm, QY = 0.95, Concentration =3.88 × 10^8^ (1/cm^3^), *η_opt_* = 22.0142 (%). (**d**) L = 10 cm, *R* = 3 cm, QY = 0.95, Concentration = 3.88 × 10^8^ (1/cm^3^), *η_opt_* = 20.1008 (%).

**Figure 18 nanomaterials-12-03594-f018:**
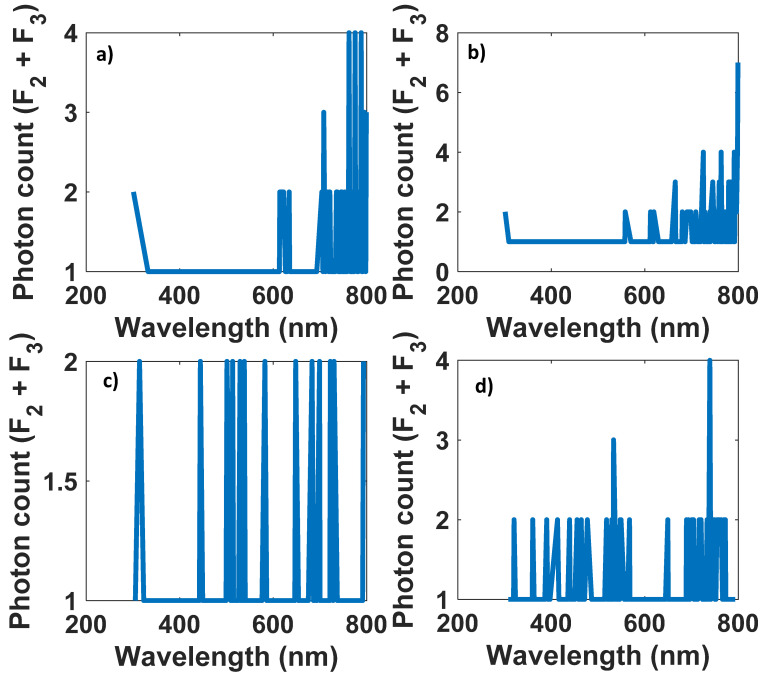
Structures with optical efficiencies lower than 1 (%). (**a**) L = 2 cm, *R* = 1 cm, QY = 0.3, Concentration = 3.88 × 10^10^ (1/cm^3^), *η_opt_* = 0.1700 (%). (**b**) L = 2 cm, *R* = 3 cm, QY = 0.3, Concentration = 3.88 × 10^10^ (1/cm^3^), *η_opt_* = 0.1852 (%).(**c**) L = 2 cm, *R* = 1 cm, QY = 0.3, Concentration = 3.88 × 10^7^ (1/cm^3^), *η_opt_* = 0.1160 (%). (**d**) L = 4 cm, *R* = 1 cm, QY = 0.3, Concentration = 3.88 × 10^7^ (1/cm^3^), *η_opt_* = 0.2084 (%).

**Table 1 nanomaterials-12-03594-t001:** Parameters related to FDTD simulation.

Parameter	Value or Type	Unit
FDTD simulation type	3D	-
Simulation time	800	fs
Temperature	300	K
FDTD region *x*, *y*, *z* span	3200	nm
FDTD background material index	1.46	-
FDTD mesh type	Custom non-uniform	-
Mesh spacing	1	nm
Boundary condition in all directions	Perfectly Matched Layer (PML)	-
Source type	Planar TFSF source	-
Source *x*, *y*, *z* span	1600	nm
Source direction	Forward	-
Source amplitude	1	-
Source wavelength range	300–800	nm
Scattering calculation *x*, *y*, *z* span	1800	nm
Absorption calculation *x*, *y*, *z* span	300	nm
Shell material	Si	-
Core material	SiO_2_	-
Shell radius (R2)	Sweeping dimensions (85–95)	nm
Core radius (R1)	6	nm

**Table 2 nanomaterials-12-03594-t002:** Optimal efficiencies for different lengths and optimal concentrations in a radius of 1 cm and quantum yields of 0.3, 0.6, and 0.95.

	Length (cm)	2	4	6	8	10
QY = 0.3	Optimal concentration (cm^−3^)	3.88 × 10^9^	3.88 × 10^8^	3.88 × 10^8^	3.88 × 10^8^	3.88 × 10^8^
	Efficiency *η_opt_* (%)	1.0180	1.2034	1.5200	1.7832	1.9844
QY = 0.6	Optimal concentration (cm^−3^)	3.88 × 10^9^	3.88 × 10^9^	3.88 × 10^9^	3.88 × 10^8^	3.88 × 10^8^
	Efficiency *η_opt_* (%)	2.5738	3.1016	3.3340	3.8666	4.3932
QY = 0.95	Optimal concentration (cm^−3^)	3.88 × 10^9^	3.88 × 10^9^	3.88 × 10^9^	3.88 × 10^9^	3.88 × 10^9^
	Efficiency *η_opt_* (%)	6.1952	7.8912	8.4874	8.8340	8.9946

**Table 3 nanomaterials-12-03594-t003:** Optimal efficiencies for different lengths and optimal concentrations in a radius of 3 cm and quantum yields of 0.3, 0.6, and 0.95.

	Length (cm)	2	4	6	8	10
QY = 0.3	Optimal concentration (cm^−3^)	3.88 × 10^8^	3.88 × 10^8^	3.88 × 10^8^	3.88 × 10^8^	3.88 × 10^8^
	Efficiency *η_opt_* (%)	2.0432	2.9740	3.5698	4.0784	4.5104
QY = 0.6	Optimal concentration (cm^−3^)	3.88 × 10^8^	3.88 × 10^8^	3.88 × 10^8^	3.88 × 10^8^	3.88 × 10^8^
	Efficiency *η_opt_* (%)	4.3908	6.5162	8.0778	9.4004	10.4696
QY = 0.95	Optimal concentration (cm^−3^)	3.88 × 10^9^	3.88 × 10^9^	3.88 × 10^8^	3.88 × 10^8^	3.88 × 10^8^
	Efficiency *η_opt_* (%)	9.4390	13.1292	15.1644	17.7336	20.1008

**Table 4 nanomaterials-12-03594-t004:** Optimal efficiencies for different lengths and optimal concentrations in a radius of 5 cm and quantum yields of 0.3, 0.6, and 0.95.

	Length(cm)	2	4	6	8	10
QY = 0.3	Optimal concentration (cm^−3^)	3.88 × 10^8^	3.88 × 10^8^	3.88 × 10^8^	3.88 × 10^8^	3.88 × 10^8^
	Efficiency *η_opt_* (%)	2.6706	4.0234	4.9420	5.5804	6.1712
QY = 0.6	Optimal concentration (cm^−3^)	3.88 × 10^8^	3.88 × 10^8^	3.88 × 10^8^	3.88 × 10^8^	3.88 × 10^8^
	Efficiency *η_opt_* (%)	5.9408	9.1848	11.3164	13.0406	14.4636
QY = 0.95	Optimal concentration (m^−3^)	3.88 × 10^8^	3.88 × 10^8^	3.88 × 10^8^	3.88 × 10^8^	3.88 × 10^8^
	Efficiency *η_opt_* (%)	10.6198	17.2854	22.0142	25.7964	29.0990

## Data Availability

Not applicable.

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
