# Peer review of "A Monte-Carlo/FDTD Study of High-Efficiency Optical Antennas for LED-Based Visible Light Communication"

_nanomaterials, 2022, doi:10.3390/nano12203594_

Round 1
Reviewer 1 Report
1. The journal Nanomaterials topics cover nanomaterials (nanoparticles, films, coatings, organic/inorganic nanocomposites, quantum dots, graphene, carbon nanotubes, etc.); Methods (synthesis, characterization, simulation, etc.). By reading carefully, I don't think this manuscript relate closely with the topic of Nanomaterials.
2. The subject of this manuscript is mainly an FDTD design of cylindrical optical antenna. Unfortunately, it is not micro-nano scale, it relate closely with the topics of optical engineering.
3. In the Abstract section, the author does not seem to present specific technical results, but seems to provide some review language.
4. In the Introduction section, the author expounds a series of basic knowledge and progress in the field of visible light communication, but the illustration of the latest technological progress and challenges is very vague.
5. In the Conclusion section, there is some qualitative results of cylindrical optical antenna, and quantitative technical conclusions are scarce.
6. Overall, the manuscript seems more like a review paper than a professional technical paper
Author Response
Dear Editor
Enclosed is the revised version of our paper entitled “A Monte-Carlo/FDTD study of high-efficiency Optical Antennas for LED-based Visible Light Communication,” submitted for your consideration and publication in this journal. We addressed all comments in the body of the paper and in the following we present a short explanation about each item.
Bests
Ali Rostami

Reviewer 2 Report
In this work, Ali Rostami et al. proposed emitting diodes based on SiO2@Si QDs as the sources in a VLC system. The results in the manuscript are reasonable and credible, but the creativity of the work may be not high enough. Thus, I recommended it to be considered in this journal after the questions below are addressed.
1. As shown in Figure 1, the luminescent peak at 450 nm was expected to origin from the emission of Si QDs. However, the Si QDs with the large sizes may not emit the short-wavelength emission, as descripted in reported literatures, such as Journal of Semiconductors, 2018, 39, 061008; Nano Energy, 2018, 54, 383.
2. There are some scattering peaks in SiO2/Si nanoparticles, and the authors should explain their origins and provide the differences of the results in Figure 11 and 12.
3. For the sources applied in the VLC, the short luminescent decay time is the requisition of the high-frequency and fast-respond communication (Photonics Research, 2022, 10, 4, 1039). However, due to the nature of indirect bandgap, the long luminescent decay time of the Si may impede their applications in VLC. Thus, the authors are recommended the reference above and explain it.
4. The English writing errors and grammar mistakes should be checked and revised.
Author Response

(The authors gave the same response as above.)
